# Rejuvenation of plasticity via deformation graining in magnesium

Bo-Yu Liu[1,10], Zhen Zhang [2,10✉], Fei Liu[1], Nan Yang[1], Bin Li[3], Peng Chen[4], Yu Wang[5], Jin-Hua Peng[6], Ju Li [7,8✉], En Ma [9✉] & Zhi-Wei Shan [1✉]

Magnesium, the lightest structural metal, usually exhibits limited ambient plasticity when compressed along its crystallographic *c*-axis (the "hard" orientation of magnesium). Here we report large plasticity in *c*-axis compression of submicron magnesium single crystal achieved by a dual-stage deformation. We show that when the plastic flow gradually strain-hardens the magnesium crystal to gigapascal level, at which point dislocation mediated plasticity is nearly exhausted, the sample instantly pancakes without fracture, accompanying a conversion of the initial single crystal into multiple grains that roughly share a common rotation axis. Atomic-scale characterization, crystallographic analyses and molecular dynamics simulations indicate that the new grains can form via transformation of pyramidal to basal planes. We categorize this grain formation as "deformation graining". The formation of new grains rejuvenates massive dislocation slip and deformation twinning to enable large plastic strains.

[1] Center for Advancing Materials Performance from the Nanoscale (CAMP-Nano) and Hysitron Applied Research Center in China (HARCC), State Key Laboratory for Mechanical Behavior of Materials, Xi'an Jiaotong University, Xi'an 710049, People's Republic of China. [2] School of Materials Science and Engineering, Hefei University of Technology, Hefei 230009, China. [3] Department of Chemical and Materials Engineering, University of Nevada, Reno, NV 89557, USA. [4] School of Materials Science and Engineering, Jilin University, Changchun 130022, P.R. China. [5] CAS Key Laboratory of Mechanical Behavior and Design of Materials, Department of Modern Mechanics, University of Science and Technology of China, Hefei 230027, PR China. [6] School of Materials Science and Engineering, Jiangsu University of Science and Technology, Zhenjiang 212003, China. [7] Department of Nuclear Science and Engineering, Massachusetts Institute of Technology, Cambridge, MA 02139, USA. [8] Department of Materials Science and Engineering, Massachusetts Institute of Technology, Cambridge, MA 02139, USA. [9] Center for Alloy Innovation and Design (CAID), State Key Laboratory for Mechanical Behavior of Materials, Xi'an Jiaotong University, Xi'an 710049, People's Republic of China. [10]These authors contributed equally: Bo-Yu Liu, Zhen Zhang. ✉email: zhenzhang.materials@gmail.com; liju@mit.edu; maen@xjtu.edu.cn; zwshan@xjtu.edu.cn

Lightweight Mg and its alloys have received tremendous attention in recent years due to their potential applications for energy savings and emission reduction[1–5]. A major drawback of Mg, in terms of mechanical properties, is the limited plasticity when compressed along its *c*-axis (the [0001] direction of hexagonal-close-packed (HCP) structure)[6]. Unfortunately, compression along this "hard" *c*-axis is frequently encountered during wrought processing of Mg, because basal texture develops during the processing, orienting most grains into *c*-axis compression[7,8]. Therefore, the plastic deformability of Mg in *c*-axis compression and the underlying mechanisms are of high interest[9–11]. The low *c*-axis plasticity can be attributed to the high critical resolved shear stress (CRSS) required to activate pyramidal dislocations[12–15], as well as to the formation of {10Ī1} contraction twins that promote strain localization and crack nucleation[16].

In this work, we discover a plastic deformation mechanism, deformation graining (DG), of Mg crystal in *c*-axis compression at high stresses. This mechanism abruptly kicks in after strain hardening to high flow stresses in submicron Mg, and transforms the single crystal into multiple grains with new orientations. The graining allows rejuvenated slip and twinning in the new grains and large plasticity is achieved.

## Results

### The two-stage plastic deformation
The present work started with in-situ *c*-axis compression tests on submicron Mg single crystals inside a transmission electron microscope (TEM). The Schmid factors for various slip and twinning systems are shown in Supplementary Table 1. Eleven samples were tested (fractured samples due to misalignment were excluded) and all of them exhibited a two-stage plastic deformation behavior that led to large strains: the pillars first underwent gradual and uniform plastic deformation mediated by pyramidal ⟨*c* + *a*⟩ dislocations (no twinning occurred), which caused pronounced strain hardening to a flow stress level over 1.0 GPa (Stage-1). These ⟨*c* + *a*⟩ dislocations are of edge, screw and mixed types, and they can glide on pyramidal I and II planes[11]. The sample then suddenly pancaked and generated a large plastic strain burst (Stage-2). In this paper, we focus on Stage-2, where an unusual mode of material response is set off to trigger large plastic flow. Atomic-scale TEM observation and molecular dynamics (MD) simulation were used to shed light on the mechanism involved.

One typical example of in-situ compression tests is shown in Fig. 1 and Supplementary Movie 1. The pillar axial direction is *c*-axis. The pillar was first uniformly deformed to a strain of ~36.7% with obvious strain hardening. The plastic deformation in this stage was mainly mediated by dislocation slip and the dislocation density continuously increased. No deformation twinning was found in this stage. The load drops in the stress-strain curve correspond to the formation and fast propagation of dislocations. After the stress reached ~1.0 GPa, a feature of new grain formed in the lower right corner of the pillar (Fig. 1c). This feature exhibits lighter contrast than its surrounding region, suggesting a lower defect density inside. The right surface near this feature slightly bulged out, as marked by the black arrow, indicating that the accompanying strain was relaxed by the free surface. Then, dark contrast emerged and spread over this region (Fig. 1d), suggesting dislocation or twinning activity inside. Soon after, the pillar suddenly pancaked with a large strain burst ~19.7%. The maximum sample height reduction was as large as ~56.4% and no cracking or fracture was observed in the flattened specimen. After this dramatic strain burst, further height reduction can be achieved when compressing was continued (an example is shown in Supplementary Fig. 1–P11 and Supplementary Movie 2). Such

large plasticity along the *c*-axis and the two-stage deformation behavior (uniform deformation followed by abrupt pancaking) was reproducible, as summarized in Supplementary Fig. 1 and Table 2. The plastic strains achieved in these submicron Mg pillars are remarkable, drastically contrasting bulk Mg single crystals which typically have *c*-axis compression strains <7%[6].

### Formation of multiple grains in the pancaked sample
Since Stage-2 deformation occurred abruptly, detailed microstructure evolution in this stage was difficult to be captured in real time. We then characterized the microstructure in the pancaked pillar *post mortem*. Surprisingly, the pancaked pillar was no longer single crystal but composed of multiple grains, as indicated by the selected area diffraction patterns (SADP) shown in Fig. 1f. The specimen was then thinned down to ~200 nm for TEM characterization (Fig. 2). According to the SADP, dark-field TEM images, and high-resolution TEM images (Supplementary Figs. 2 and 3), we found that the flattened sample was composed of 13 grains with various sizes and shapes. These grains have large misorientation angles. Two low-angle grain boundaries can be seen within grain 10 (Supplementary Fig. 3b), which is a manifestation of dislocation activities inside the new grains, or an indicator that grain 10 is composed of three grains that underwent similar transformation routes then merged. The *c*-axis of each grain is marked by the white arrows (Fig. 2a). Dark-field TEM images using different **g** vectors show that high density of basal ⟨*a*⟩ dislocations (Fig. 2b) and non-basal, likely ⟨*c* + *a*⟩ dislocations (Fig. 2c), indicating a rejuvenation of dislocation activities in the newly formed grain.

The zone axis of all the grains is ~⟨2Ī̄10⟩, i.e. the *a*-axis of HCP unit cell, suggesting that these grains share a common ⟨2Ī̄10⟩ axis. The orientation relationships between adjacent grains can be classified into four groups, which are close to 62°⟨2Ī̄10⟩, 32°⟨2Ī̄10⟩, 24°⟨2Ī̄10⟩, and 86°⟨2Ī̄10⟩, as summarized in Supplementary Table 3. Except for those close to 86.3°⟨2Ī̄10⟩ that are generated by {10Ī2} twinning, other orientation relationships cannot be simply classified as deformation twinning. Such co-axial and misorientation relationship were reproducible, another example is shown in Supplementary Fig. 4. High-resolution TEM images show that there are numerous {0002}/{10Ī0}, {0002}/{10Ī1}, {0002}/{10Ī3}, {10Ī0}/{10Ī1}, and {10Ī0}/{10Ī3} interfaces on the grain boundaries (Supplementary Fig. 3).

### Growth/shrink of a new grain during loading/unloading
The high stress level, rapid shape-change, and formation of new grains with specific orientation relationships and boundary structures indicate that a new plasticity mechanism is activated in Stage-2 deformation. Although it was difficult to record the detailed microstructure evolution during the abrupt pancaking, in-situ movies provide insight into how these new grains formed inside the parent single crystal. As shown in Fig. 1c, a feature in the form of a new grain appeared prior to the onset of the abrupt pancaking. In another pillar that was undergoing *c*-axis compression, as shown in Fig. 3, we found a similar feature of a new grain, which formed and grew when the stress reached ~1.0 GPa during loading, shrank during unloading, and grew again upon reloading, indicating that its boundaries were highly "glissile" at room temperature. Some segments of these boundaries were parallel to the traces of {10Ī1} planes of the matrix, suggesting that a mobile boundary along {10Ī1} plays an important role in the formation and growth of this feature.

### Atomic structure of the grain boundaries
To better characterize the new grain formation, in another test, we interrupted the deformation and retracted the diamond punch, when a similar

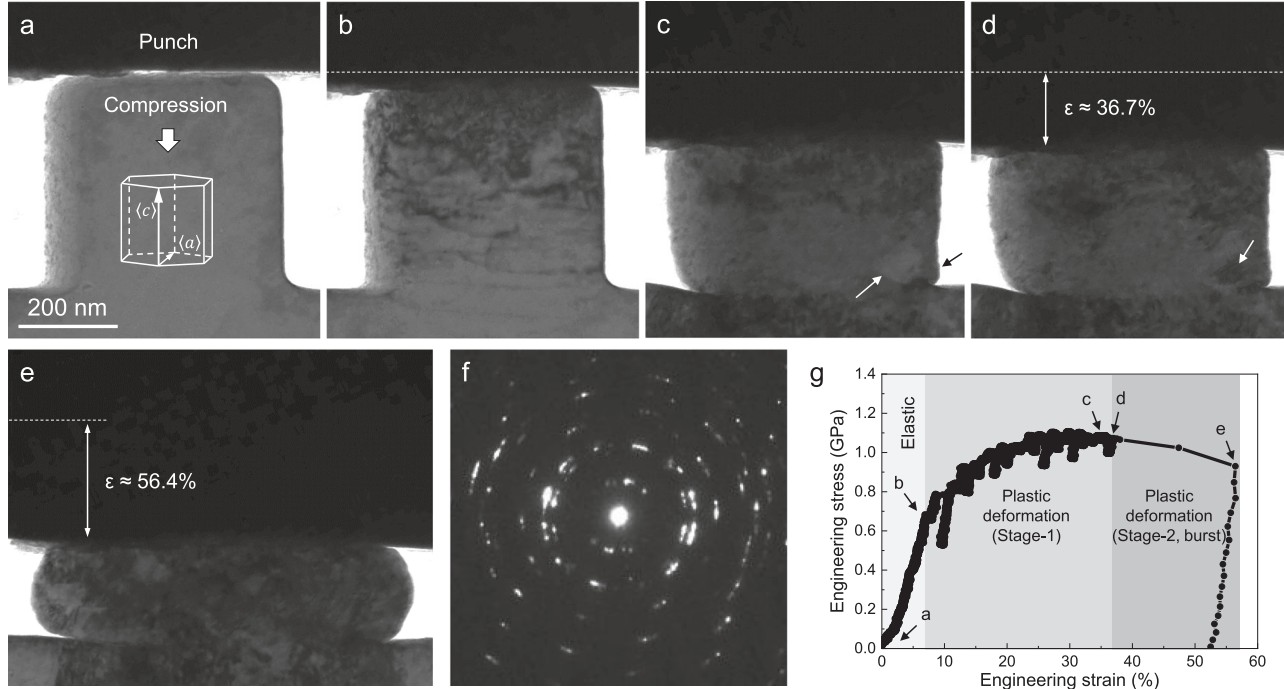

**Fig. 1 A submicron Mg single crystal pillar uniformly deformed under *c*-axis compression, then flattened in a strain burst. a** The initial pillar before compression. Inset hexagonal unit cell shows the loading orientation. Viewing direction, ∼⟨2$\bar{1}\bar{1}$0⟩. **b** Formation and motion of dislocations. **c** A new grain formed in the lower right corner of the pillar (white arrow). **d** Dark contrast emerged in the new grain suggesting dislocation or twinning activity inside. **e** The pillar suddenly flattened. **f** Selected area diffraction patterns acquired from the flattened sample. **g** Corresponding stress-strain curve showing the two-stage plastic deformation.

new grain emerged in the view. The strain achieved was ~26.5% (P10 in Supplementary Fig. 1). From the observed critical strain just before the strain burst in Stage-2 in the present work (~22–39%, Supplementary Table 2), we can infer that this pillar was strained to near the end of Stage-1. The dark-field TEM image and SADP confirmed the formation of a new grain (Fig. 4a). The new grain had a polygonal shape with dimensions around 250 nm. It shared the ⟨2$\bar{1}\bar{1}$0⟩ with the matrix, similar to the co-axial relation displayed in Fig. 1. A ~62°⟨2$\bar{1}\bar{1}$0⟩ orientation relationship exists between the matrix and the new grain. In the SADP, the {0002} reflection almost overlaps with a set of {10$\bar{1}$1} reflections, indicating a parallel relation between the basal plane in the new grain and the {10$\bar{1}$1} pyramidal plane in the matrix, and vice versa. Moreover, a number of {10$\bar{1}$1}/{0002} interfaces, i.e. pyramidal/basal (designated as Py/B hereafter) interfaces, were observed along the grain boundaries (Fig. 4b, c). A one-to-one atomic correspondence across the Py/B interface can also be observed (Fig. 4d). Similar boundary structure was observed in another test, where we captured a new grain with a much smaller size, ~50 nm, which can be regarded as a nucleus (Fig. 5). Py/B and {10$\bar{1}$0}/{10$\bar{1}$3} interfaces can be seen on the boundary.

MD simulation was then performed to help reveal the mechanism for new grain formation (Supplementary Fig. 5). A pure Mg single crystal was compressed along its *c*-axis. During compression, 4 new grains formed. These new grains and the matrix shared a common ⟨2$\bar{1}\bar{1}$0⟩ axis and the orientation relationship is ~62°⟨2$\bar{1}\bar{1}$0⟩. The basal plane in the new grain was parallel to the pyramidal plane in the matrix. Such orientation relationship is consistent with our experimental observations. Multiple Py/B interfaces were observed on the grain boundaries. Interestingly, the Py/B interfaces are mobile, growing the new grain. Comparison between the {10$\bar{1}$1} and {0002} planes shows that the atomic structure of {10$\bar{1}$1} plane is

very close to that of {0002} (Fig. 6), suggesting that the two planes are inter-transformable, rendering the interface highly mobile.

**Pyramidal-basal transformation.** Based on the experimental observations, MD simulation and crystallographic analyses, we propose that the formation of the new grains shown in Figs. 4 and 5 is accomplished through pyramidal-basal (Py-B) transformation. In this process, atoms on an original pyramidal plane of the matrix rearrange to form the basal plane of the new grain. The high density of pyramidal ⟨*c* + *a*⟩ dislocations formed in Stage-1 deformation may facilitate such transformation, probably by displacing the atoms on pyramidal planes or by providing nucleation sites. These ⟨*c* + *a*⟩ dislocations can also serve for strain compatibility because ⟨*c* + *a*⟩ slip system can provide five independent slip modes sufficient to meet the von Mises criterion, which allows the crystal surrounding the new grain to undergo a homogeneous deformation.

The Py-B transformation produces a 62°⟨2$\bar{1}\bar{1}$0⟩ misorientation, as illustrated in Fig. 6. This 62° reorientation, combined with {10$\bar{1}$2} twinning, can well explain the origin of the grain boundary structures and misorientations. Figure 7 demonstrates a possible pathway to the formation of grains 3 and 10 shown in Fig. 2a. Grain 1 is the matrix. Part of it transforms into grain 3 via Py-B transformation. Meanwhile, the {10$\bar{1}$0}$_{grain1}$ transforms into {10$\bar{1}$3}$_{grain3}$, forming a {10$\bar{1}$0}$_{grain1}$/{10$\bar{1}$3}$_{grain3}$ interface. Part of grain 3 then transforms into grain 10 via {10$\bar{1}$2} twinning, forming the basal/prismatic ({0002}/{10$\bar{1}$0}) interface. In this process, the {0002}$_{grain1}$ transforms to {10$\bar{1}$1}$_{grain3}$ via Py-B transformation, then to {10$\bar{1}$3}$_{grain10}$ via twinning, forming {0002}$_{grain1}$/{10$\bar{1}$3}$_{grain10}$ interfaces. These plane-to-plane correspondences are schematically shown in Fig. 7b. Figure 7c shows a {10$\bar{1}$1}$_{grain1}$/{10$\bar{1}$0}$_{grain10}$ interface (the corresponding planes are delineated by blue dashed lines in Fig. 7b), which corroborates this transformation sequence. Pathways to the

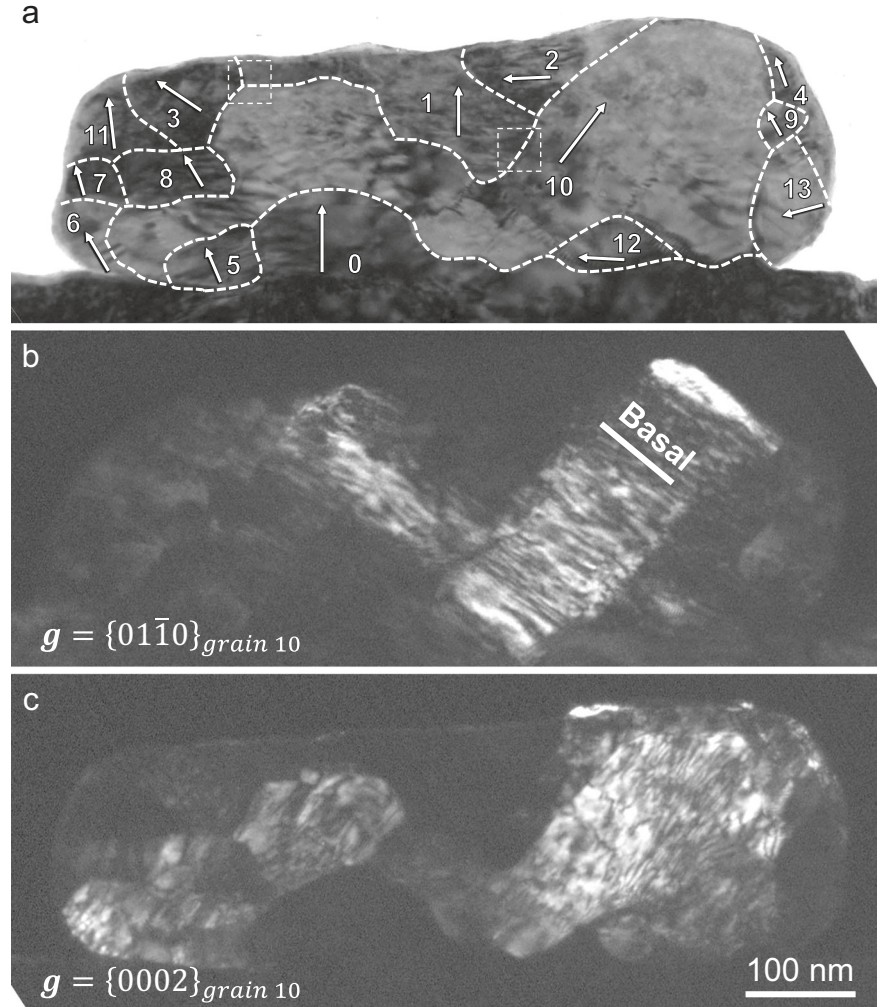

**Fig. 2 The newly formed grains contain high-density dislocations. a** The initial single crystal became 13 grains. The grain boundaries are outlined by white dashed lines. The white arrow indicates the *c*-axis of each grain. **b** High density basal dislocations in grain 10. When using $\vec{g} = \{01\bar{1}0\}$, dislocations with Burgers vector that contains $\langle a \rangle$ component are visible. These dislocations are found to generally follow the basal plane of grain 10 (marked by a white line), and they should be basal $\langle a \rangle$ dislocations. **c** High density of non-basal dislocations in grain 10. When using $\vec{g} = \{0002\}$, dislocations with Burgers vector that contains $\langle c \rangle$ component are visible. These dislocations are found to be out of basal plane, they should be non-basal $\langle c \rangle$ or $\langle c + a \rangle$ dislocations. Viewing direction, $\sim\langle 2\bar{1}\bar{1}0 \rangle$.

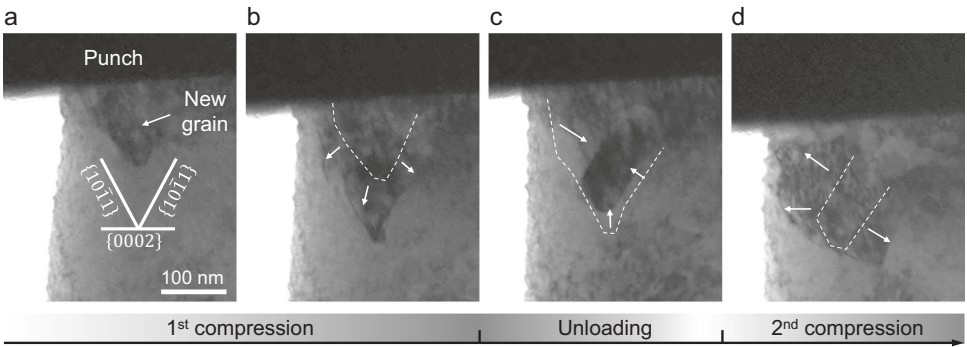

**Fig. 3 The growth and shrinking of a new grain.** This grain formed when the stress reached ~1.0 GPa at the end of Stage-1, near the contact interface between the pillar top and diamond punch. **a**, **b** It grew during loading, **c** shrank during unloading, **d** then grew again upon reloading. Dashed line marks the previous position of the boundary. The viewing direction is $\sim\langle 2\bar{1}\bar{1}0 \rangle$. The traces of $\{0002\}$ and $\{10\bar{1}1\}$ are outlined.

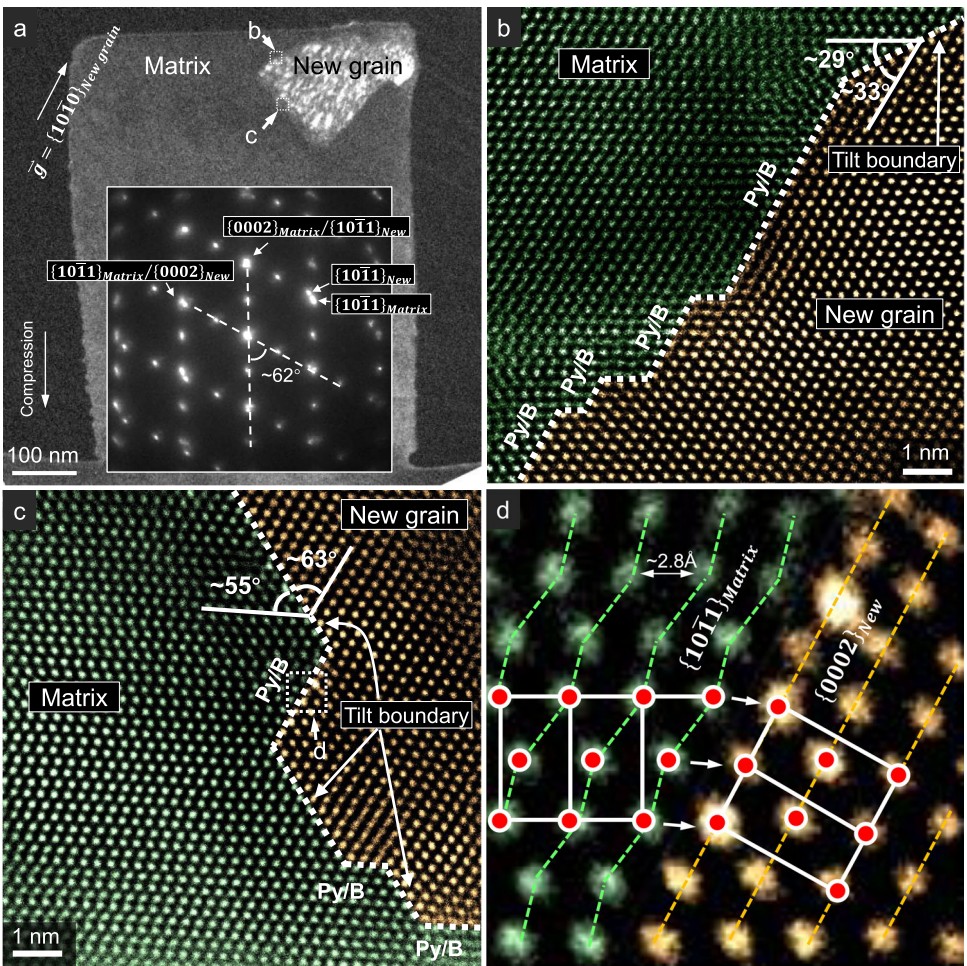

**Fig. 4 Atomic structure of the grain boundaries of a new grain formed during *c*-axis compression. a** TEM dark-field image showing the morphology of this new grain. The inset selected area diffraction patterns indicate the orientation relationship of $\{10\bar{1}1\}_{Matrix} \parallel \{0002\}_{New}$ and $\{0002\}_{Matrix} \parallel \{10\bar{1}1\}_{New}$. **b**, **c** Py/B interfaces and basal plane tilt boundaries, acquired from the places marked by the white dashed frames labeled by "**b**" and "**c**" in **a**, respectively. **d** Fine structure of the Py/B interface, acquired from the place marked by the white dashed frame labeled by "**d**" in **c**. The green and orange dashed lines are the traces of corrugated $\{10\bar{1}1\}$ and flat $\{0002\}$ planes, respectively. The HCP unit cells of the matrix and the new grain are highlighted by the white lines and red circles. The zone axis is $\sim\langle 2\bar{1}\bar{1}0\rangle$.

formation of new grains in Fig. 2a and Supplementary Fig. 3 through Py-B transformation and deformation twinning are analyzed in Supplementary Table 3. All the interfaces observed in Supplementary Fig. 3 can be explained by the above transformation process.

The mobility of Py/B interfaces, on the other hand, explains well the observed back and forth migration of the boundary during loading and unloading (Fig. 3). Such stress-driven motion belongs to the "military transformation" category, the same as martensitic transformations, deformation twinning, etc., which may involve both affine transformation and non-affine shuffling atomic displacements[17], but highly deterministic within a single domain.

## Discussion

We next discuss the nature of the observed stress-induced new grain formation. Textbook grain dynamics models based on linear-response theory such as curvature-driven von Neumann–Mullins grain growth[18] did not consider highly non-linear stress-actuated "convective" effects (such as collective grain rotation). However, modern treatments have revealed a host of unexpected phenomena[19,20] with significant stress/strain accumulation or relaxation associated with local grain boundary

motion. In particular, in the high-temperature limit, grain boundary motion is expected to become more diffusive and linear-response like (meaning if the thermodynamic driving force $\sigma$ is doubled, the speed $v$ will also double) as in the classic curvature-driven growth and Nabarro-Herring/Coble creep models; whereas as the temperature is lowered, the grain boundary motion will become more stress and strain or shear-coupled, "glissile", and demonstrate a much more thresholded behavior with respect to the thermodynamic driving force, that is, almost not moving if the driving force is below a critical value, but moving with the extreme speed if the driving force goes above the threshold ($v \propto \sigma^n$ with a large $n$). According to the dis-connection model of grain boundary motion[20], significant stress locking can develop at lower temperatures that jams the grain dynamics. But once the stress threshold is reached, new grain formation can bring in significant strain relaxation suddenly at or below room temperature ($T_{room}/T_{melt}^{Mg} = 0.32$).

The stress-induced new grain formation observed in this work is fundamentally different from dynamic recrystallization (DRX), as follows. (1) DRX typically occurs at elevated temperatures, driven by free energy reduction by the removal of stored dis-locations. Nucleation and growth of recrystallized grains are associated with thermally activated diffusion with low activation

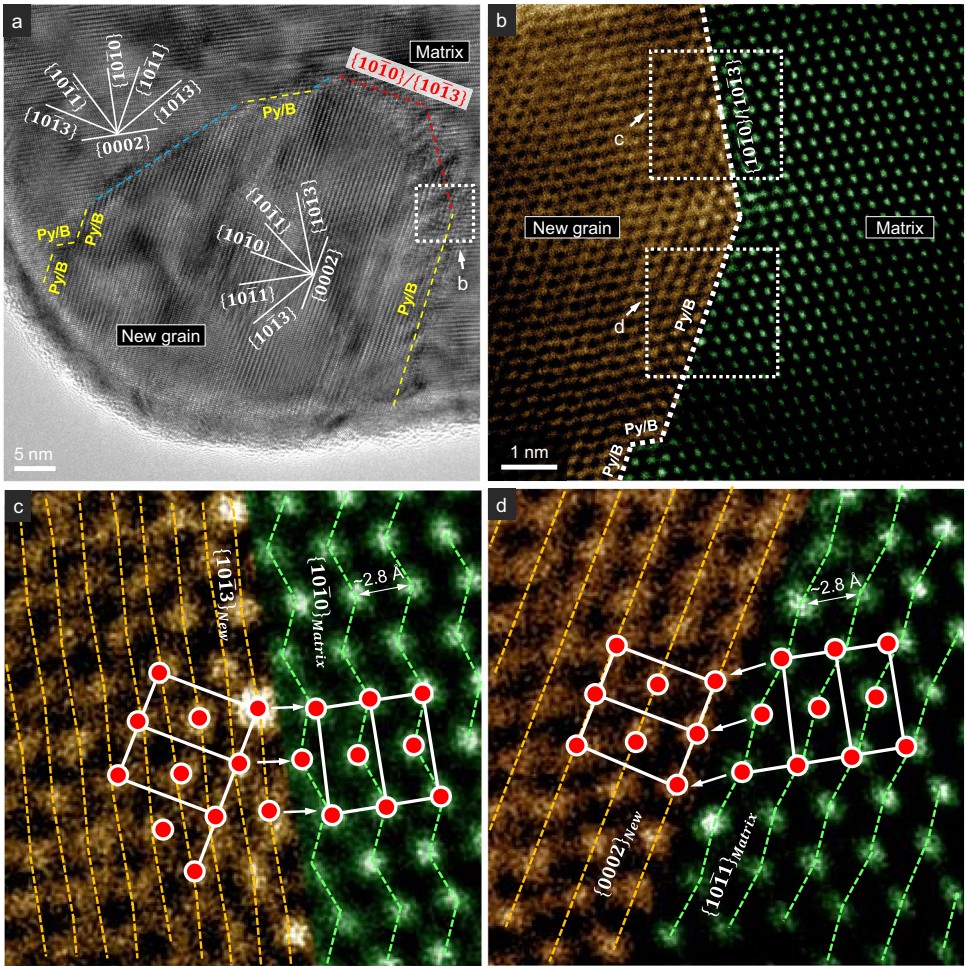

**Fig. 5 The boundary structure of a nucleus of a nano-sized new grain formed during *c*-axis compression. a** High-resolution TEM images showing that the grain boundary is composed of Py/B (the yellow dashed lines) and $\{10\bar{1}0\}/\{10\bar{1}3\}$ (the red dashed lines) interfaces, and $\{10\bar{1}3\}_{matrix}/\{10\bar{1}3\}_{new}$ boundary (the cyan dashed lines). **b** A segment that contains Py/B and $\{10\bar{1}0\}/\{10\bar{1}3\}$ interfaces, acquired from the place marked by the white dashed frame labeled by '**b**' in **a**. **c** Fine structure of the $\{10\bar{1}0\}/\{10\bar{1}3\}$ interface. The green and orange dashed lines are the traces of corrugated $\{10\bar{1}0\}$ and $\{10\bar{1}3\}$ planes, respectively. **d** Fine structure of the Py/B interface. The green and orange dashed lines are the traces of corrugated $\{10\bar{1}1\}$ and flat $\{0002\}$ planes, respectively. The HCP unit cells of the matrix and the new grain are highlighted by the white lines and red circles. The **c** and **d** are acquired from the places marked by the white dashed frames labeled by '**c**' and '**d**' in **b**, respectively. Zone axis, $\sim\langle2\bar{1}\bar{1}0\rangle$.

volume. In contrast, our experiments were conducted at room temperature. We also conducted interrupted tests to minimize the effect of possible heat accumulation generated by plastic deformation (Supplementary Fig. 1). In these tests, the two-stage plastic deformation and the abrupt pancaking also occurred, suggesting that thermally activated "civilian" diffusion was not the main driving factor for the formation of new grains. (2) DRX is strongly time and temperature dependent. At a given temperature, the growth kinetics of conventional DRX grains is proportional to $t^n$ ($t$ is time, $n = 0.3\sim0.5$)[21]. In our case, new grains form fast, even in an explosive fashion above a threshold stress, resulting in a sudden strain burst that is characteristic of "military" transformations. (3) In our experiments, a lattice correspondence exists for the grain boundary. The mobility of the grain boundaries comes from the structural similarity between the corresponding atomic planes across the boundary. The observed deformation-induced grain boundary motion is governed by short-range atomic rearrangements, different from that mediated by diffusion in conventional DRX. We call it "deformation graining" (DG) to categorize this type of nucleation and growth of grains. DRX is more prevalent at higher temperatures and lower stresses (thermally driven), whereas DG is more prevalent

at lower temperatures and higher stresses (stress driven). In DG, the migration of grain boundaries occurs via displacive movements of atoms at the interface. Shear coupled stress-driven grain boundary migration[22] and the basal/prismatic interface migration[23–25] can be considered as other examples of DG.

In the present work, the grain formation and boundary migration are proposed to be accomplished through the mechanical-loading induced Py-B transformation ("military" displacement of atoms from one crystallographic plane to another as indicated by the lattice correspondence). However, it is worth noting that, in a more general case, mechanical loading not only produces affine displacements of atoms, but also can produce non-affine displacements. These non-affine atomic displacements consist of domain-uniform "shuffling" that is of identical pattern from unit-cell to unit-cell (in the same transformation domain), and may also plus individually randomized motion (analogy with "civilian" movement). The latter, which is called deformation-induced diffusion[26], has been found to contribute to the crystallization of metallic glasses under cyclic loading well below the glass transition temperature[27]. At this point, while we can rigorously measure the affine transformation strain aspect, we cannot ascertain the ratio of domain-uniform "shuffling" versus

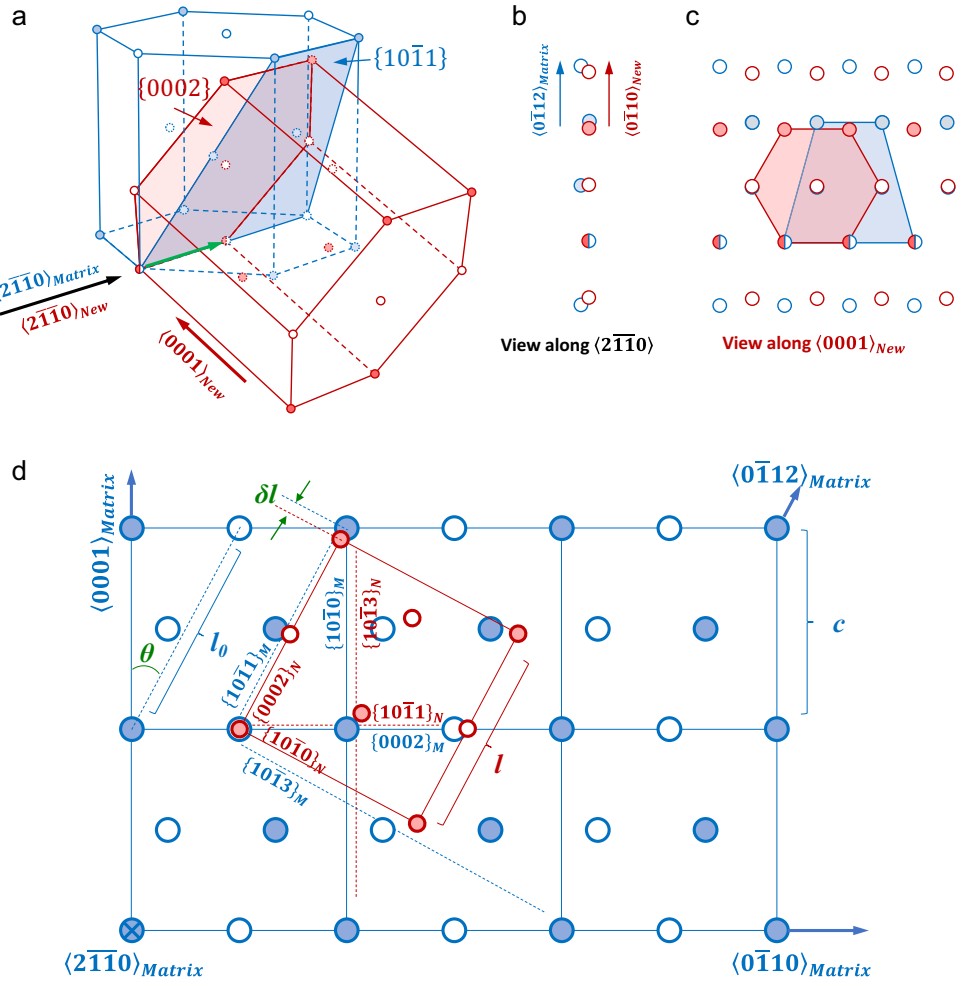

**Fig. 6 Comparison between the structure of {10$\bar{1}$1} and {0002} plane in Mg. a** A matrix (blue) and a new HCP unit cell (red) with a 62°⟨2$\bar{1}$10⟩ misorientation. Atoms on different {01$\bar{1}$0} layers are represented by solid and open circles. The two planes joining at the interface, i.e., {10$\bar{1}$1}$_{Matrix}$ and {0002}$_{New}$, are highlighted. The green arrow denotes the ⟨2$\bar{1}$10⟩ shared by the two HCP unit cells. **b, c** The projection views of the overlapped {10$\bar{1}$1}$_{Matrix}$ and {0002}$_{New}$. **d** A new HCP unit cell (in red) is delineated inside the matrix. The two lattices satisfy {10$\bar{1}$1}$_{Matrix}$ ∥ {0002}$_{New}$, {0002}$_{Matrix}$ ∥ {10$\bar{1}$1}$_{New}$, {10$\bar{1}$0}$_{Matrix}$ ∥ {10$\bar{1}$3}$_{New}$, and {10$\bar{1}$3}$_{Matrix}$ ∥ {10$\bar{1}$0}$_{New}$, with a zone axis of ⟨2$\bar{1}$10⟩.

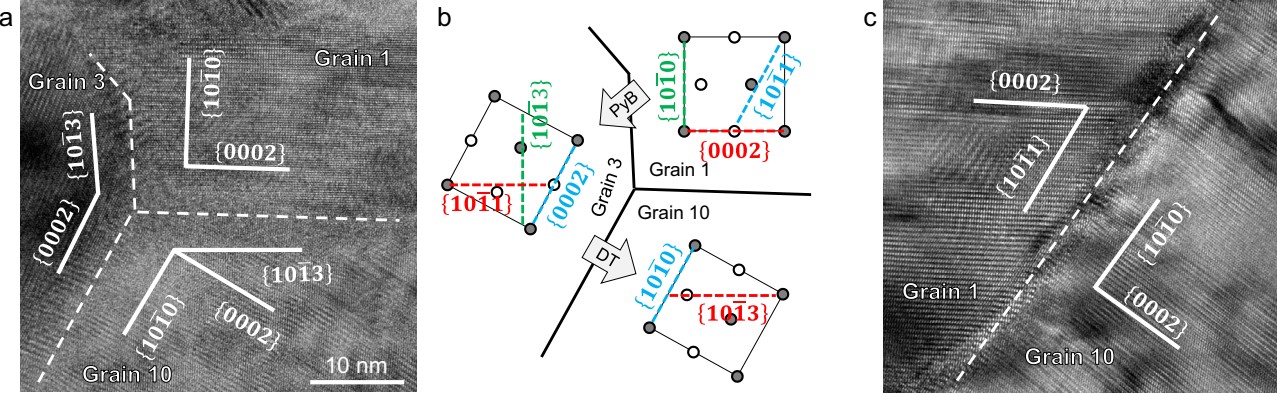

**Fig. 7 Activation of {10$\bar{1}$2} twinning following Py-B transformation. a** A triple junction of the grain boundaries between grains 1, 3, and 10. This junction is composed of {10$\bar{1}$0}$_{grain1}$/{10$\bar{1}$3}$_{grain3}$ interface, {0002}$_{grain3}$/{10$\bar{1}$0}$_{grain10}$ interface (basal/prismatic interface) and {0002}$_{grain1}$/{10$\bar{1}$3}$_{grain10}$ interface. **b** A possible pathway of the formation of grains 3 and 10 from grain 1. **c** A {10$\bar{1}$1}$_{grain1}$/{10$\bar{1}$0}$_{grain10}$ interface between grain 1 and 10, corroborating the pathways in **b**. Zone axis, ∼⟨2$\bar{1}$10⟩.

"civilian" motion, but we expect that the actual grain nucleation and growth in many situations probably be a mixture of the two[20], i.e. hybrid diffusive-displacive mode, at room temperature. We note that at the atomic level, the essential difference between a "civilian" and a "military" process is that, while the atomic registries (i.e. nearest-neighbor relationships) can change in both types of processes, changes in the former occur in a highly individually randomized manner, while in the latter in a much more collective, domain-uniform manner as reflected quantitatively by the larger activation volume[28].

As a less noticed cousin to deformation twinning and dislocation slip, DG promotes plasticity at lower temperatures and higher stresses without phase transformations. First, the deformation-induced grain formation per se can produce plastic strain (e.g. disconnection glide/climb model[20], motion of pyramidal/basal interfaces and basal/prismatic interfaces[23], and shear-coupled grain boundary migration[29]). The transformation strain induced by Py-B transformation is calculated to be ~5.85% from $\delta l \cdot \cos\theta/c$ ($\delta l$, $c$, $\theta$ are marked in Fig. 6). DG occurring via Py-B transformation is then expected to contribute to plastic strain, as do deformation twinning or martensitic transformation. The glissile boundaries as well as its capability of reversible migration can also relax stress concentration that is usually accumulated at immobile grain boundaries, thus delaying damage and cracking. Note that although the boundary migration is reversible, the total shape-change and microstructure evolution (caused by DG and subsequent dislocation and twinning activities in the new grains) are irreversible. The Py-B transformation shortens the Mg crystal along the $c$-axis of the matrix crystal, which is why this mechanism is favored under $c$-axis compression. In contrast, basal-prismatic transformation elongates the Mg crystal along the $c$-axis of the matrix and thus occurs under $c$-axis tension. In both cases, the activated deformation mode carries the plastic strain imposed by the applied load.

Second, DG facilitates the activation of basal slip in the new grains that are no longer in the same crystallographic orientation as the parent single crystal. Before DG, the dominant deformation mode is the hard $\langle c + a \rangle$ slip in $c$-axis compression. After DG, new grains form with their $c$-axes deviating from the loading direction, such that easy basal slip can be activated. Note that before DG the flow stress has already reached 1.0 GPa level, much higher than the CRSS for basal slip in submicron Mg (0.1 GPa[30]). Once new grains form, the nearly exhausted dislocation plasticity of the heavily deformed single crystal is rejuvenated (as suggested in Fig. 2 and Supplementary Fig. 1–P11). In addition to dislocation slips, deformation twinning can also activate in the new grains to accommodate plastic strain (Fig. 7 and Supplementary Table 3). Considering that the strain produced by Py-B transformation per se is relatively small, the dislocation and twinning induced plastic strain in the new grains should be the main contributor to the Stage-2 deformation (Supplementary Table 4). In our experiments, the stress level is very high. As a result, many new grains may simultaneously form in such an overloaded sample and, profuse dislocations and twins can be immediately activated once these new grains form, leading to a fast and large deformation (strain burst). Dislocation slip can also help to satisfy strain compatibility at grain boundaries when the new grains grow to impinge on each other. Moreover, in our experiments, formation of new grains by Py-B transformation "rotates" the crystal around $\langle 2\bar{1}\bar{1}0 \rangle$ but cannot provide strain along this "rotation" axis, that is, the deformation is anisotropic, as shown by the top view of the flattened sample P8 in Supplementary Fig. 1. As a result, dislocation slip is needed to make the deformation more homogeneous.

Third, $\{10\bar{1}\bar{1}\}$ contraction twin was not observed in our experiments. Similarly, previous works also found that

contraction twin is suppressed in $c$-axis compression tests on micro- and nano- scale Mg single-crystal pillars[11,31,32]. The absence of contraction twins would contribute to the improved plasticity because this type of twins usually has a sharp and narrow shape thus can serve as crack initiation sites in Mg[16]. The differences between Py-B transformation and $\{10\bar{1}1\}$ twinning are summarized in Supplementary Fig. 6 in terms of crystallography. Note that although contraction twinning does not occur in our tests, the phenomena that massive dislocation slip activated in the new grains, as well as the grain refinement by DG, are similar to the activation of basal slip in the contraction twins (twinning can reorient the lattice) and grain refinement by twinning induced DRX[33–35]. In addition, intergranular plasticity (e.g. grain boundary sliding[36]) is possibly another flow mechanism that may be activated after new grain formation.

In summary, the present work reveals a phenomenon, "deformation graining", which has the potential of enhancing the plasticity of Mg under proper activating conditions. Whether this is viable in bulk materials remains to be explored, but there are hints in the literature that DG may have played a role in some unusual observations. One case is some unusual misorientation relationships found in deformed Mg alloy, which cannot be unambiguously classified in terms of known twinning systems[37,38], suggesting other potential DG modes. Another is the significantly improved roll-ability when Mg alloys were subjected to "fast rolling"[39], where the imposed stress is expected to be much higher than that under normal rolling speeds. DG may have been enabled under higher-than-normal rolling strains in this case. Our results may also have another implication: if Mg (and possibly other HCP metals) is processed under high-stress conditions (such as in samples of small dimensions or made of nanocrystalline grains, or under high strain rates like fast rolling or impact conditions), the normally poor $c$-axis plasticity could be overcome through the activation of additional plastic deformation modes, which revive dislocation activities that are otherwise exhausted, even when the imposed loading starts out in the hardest orientation. In general, to what extent the DG mechanism can be activated in bulk materials, as well as its possible effects on mechanical properties, microstructure evolution, and processability warrant further exploration.

## Methods

**Specimen fabrication**. The pure Mg single crystal was produced by Bridgman method using graphite crucible under the protection of 99.99% Ar atmosphere. The submicron pillar was fabricated by using FIB milling from the bulk scale material. The cross-section of the pillar was in square shape, with a width and thickness of ~400 nm. Three aspect ratios were used: 3:1, 2:1, and 1:1. Using low aspect ratio can avoid premature failure that is usually caused by misalignment between punch and pillar. The height direction of the pillar was set to be $\langle 0001 \rangle$. The thickness direction was set to be $\langle 2\bar{1}\bar{1}0 \rangle$.

**In-situ tests and TEM characterization**. The in-situ uniaxial compression test on submicron pillar was performed on a Hysitron PicoIndenter (PI95) inside a JEOL 2100F TEM (200 keV). The strain rate was in the level of $10^{-2}\,\text{s}^{-1}$. The TEM view direction was parallel to the pillar thickness, i.e. $\langle 2\bar{1}\bar{1}0 \rangle$. The in-situ microstructure evolution was recorded by using a GATAN 833 camera. The atomic scale imaging was carried out on a spherical aberration-corrected STEM (ARM-200).

**MD simulation**. Atomistic simulation of $c$-axis compression of a Mg single crystal was performed by using embedded atom method (EAM) interatomic potentials developed by Sun et al[40]. Large-scale atomic/molecular massively parallel simulator (LAMMPS) was used[41]. A perfect single crystal containing 2,112,000 atoms was constructed and a compressive load was applied along the $c$-axis. Free surfaces were applied to all three dimensions. A strain rate of $1.9 \times 10^8$/s was generated by moving the atoms on the right-side free surface. The temperature was maintained at 100 K.

**Estimation of strain-component**. The strain produced by different plastic carriers in the pillar shown in Fig. 2 is estimated by the following procedure. (1) The strain produced by Py-B transformation and subsequent $\{10\bar{1}2\}$ twinning of each grain,

$\varepsilon_{Py-B} = f_1 \times 5.85\%$, $\varepsilon_{twin} = f_2 \times 6.28\%$, where $f_1$ is the volume fraction of the grains that experienced Py-B transformation, and $f_2$ the volume fraction of {10$\bar{1}$2} twins, 5.85% and 6.28% are the plastic strains produced by Py-B transformation and {10$\bar{1}$2} twinning, respectively. Because the image acquired by TEM is a two-dimensional projection, the volume (area × thickness) fraction was approximated by the "area fraction". This approximation is generally accurate in the current experiments because the strain produced along the direction of pillar thickness is small. (2) The strain produced by dislocation slip in the new grains, $\varepsilon_{dislocation} = \varepsilon_{Stage-2} - \sum \varepsilon_{Py-B} - \sum \varepsilon_{twin}$, where $\varepsilon_{Stage-2}$ is the total strain produced in Stage-2 deformation, $\sum \varepsilon_{Py-B}$ and $\sum \varepsilon_{twin}$ are the strain produced by Py-B transformation and twinning of all grains. The calculation results are shown in Supplementary Table 4. This method of strain-component analysis was developed by a previous work[42].

## Data availability

All data generated or analyzed during this study are included in this published article (and its supplementary information files).

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

## Acknowledgements

We acknowledge the supports by the National Natural Science Foundation of China (51971168, 52031011, 52022076, and 51871084) and National 111 Project 2.0 (BP0618008). B.L. thanks support from U.S. National Science Foundation (CMMI-1635088, 2016263, and 2032483). E.M. acknowledges XJTU for hosting his research at Center for Alloy Innovation and Design. We thank Dr. L. Lu, Dr. C.W. Guo, Dr. P. Zhang, Dr. Y.B. Qin, D.L. Zhang, and J. Xiao for assistance in sample preparation and TEM experiments, and Dr. W.B. Jiang (Chinese Academy of Sciences) for providing Mg single crystals.

## Author contributions

Z.W.S., Z.Z., J.L., and E.M. designed and supervised the project. B.Y.L., Z.Z., F.L., N.Y., and J.H.P. carried out the experiments and analyzed the data. B.L., Z.Z., P.C., and Y.W. carried out the MD simulations and interpreted the simulation results. B.Y.L., E.M., J.L., Z.W.S., B.L., and Z.Z. wrote the paper. All authors contributed to data analysis and the discussion of the results.

## Competing interests

The authors declare no competing interests.

**Additional information**

