## [Peer Review File · Nature Communications]

Rejuvenation of plasticity via deformation graining in magnesiumREVIEWER COMMENTS

Reviewer #1 (Remarks to the Author):

Rejuvenation of plasticity by deformation graining in magnesium

In this work, the authors investigated the deformation behavior of pure Mg micropillars by in-situ TEM analysis and MD simulation. The authors found a huge burst in on the strain-strain curves with an interesting pancaked deformation on the specimen geometry. Through the in-situ TEM observation and MD simulation, deformation graining by rearranging of atoms on the pyramidal plane to the basal plane, which contributes to generating new grains, is proposed. This mechanism contributes be to increasing the ductility of the material by decreasing the previous dislocations and activation of new slips in newly generated grains. It is a novel and interesting topic.

After reading the manuscript carefully, in the reviewer's opinion, the outcomes in the current work show sufficient novelty and the scientific clarity to merit publication in Nature communication after a revision. Given below are concerns and questions raised with respect to the key findings in the work.

- 1) As the authors mentioned it is not possible to catch the deformation in stage II due to the time resolution limits in this stage. The reviewer suggests showing the orientation of the 'matrix' at the end of the homogeneous deformation (e.g. at point d in Figure 1 on the strain-stress curve) in order to exclude the possibility of double twinning and other possible twinning mechanisms.
- 2) In the reviewer's opinion, the nucleation mechanism of the Pyr/B transformation or the new grains remains unclear. According to the supplementary video, between b and c (Figure 1), the total dislocation density decreased sharply based on the contrast. Either the dislocations escape from the surface or dynamic recovery occurs simultaneously. Therefore, again the reviewer recommends the authors investigate the sample before pancaking in detail, including the dislocation density distributions, dislocation type and the orientation of the material. In this way, we could have a clear understanding of the generation mechanism of those new grains better.
- 3) In Figure 3, to exclude the possibility of twinning and detwinning, the orientation of the new grains should be also given.
- 4) In Figures 2 and 5, according to the authors' scenario (Page 5 Line 4-18), the matrix will firstly transform to an orientation of grain 3 by Pyr/B and then twined to grain 10. This indicates that the matrix should firstly be transformed to grain 3 nearly through the whole specimen dimension and then more than 80% of it twinned to grain 10. Is the Pyr/B transformation possible to happen all over the specimen? In addition, since the CRSS of tension twin is so low in Mg, under so high stress why grain 3 is not totally twinned? The reviewer is not convinced. Further explanations are necessary.

5) Page 8 Line 4, the author cited as an example of high ductility in Mg single crystal at room temperature (Ref. 28). By carefully read this paper, from the reviewer's understanding, the author compressed the crystal along $\langle 11\text{-}20 \rangle$ direction, where tension twin is preferential and the new grains come out mostly by dynamic recrystallization at room temperature. This example here could probably mislead the readers about the difference between DG, twinning and DRX.

6) It would be very interesting if we could find this new mechanism in the bulk material since the size effect plays a crucial role in the mechanical response in such a small scale.

Reviewer #2 (Remarks to the Author):

The manuscript present results of nanopillar experiments on Mg single crystal deformed along the c-axis in compression at room temperature. As the pillar is compressed, plastic flow due to non basal dislocation slip (clearly visible on the video) gradually strain hardens the single crystal up to gigapascal level, at which new grain is formed and then the crystal suddenly deforms plastically $\sim 20\%$, while the applied load decreases without fracture. The accompanying video very nicely documents this unusual deformation process. After this large compressive deformation and unloading, the single crystal becomes decomposed into multiple small grains that share a common rotation a-axis of the HCP lattice. The new microstructure consisting of these multiple small grains within the deformed single crystal pillar is carefully analyzed by TEM a HRTEM as concerns the lattice rotation and type of new interfaces. The deformation process is claimed to represent new deformation mechanism of magnesium called "deformation graining" activated at low temperatures and high stresses. It is suggested that this new deformation mechanism can be possibly activated in cold working of magnesium.

The authors claim that the observed lattice rotations are produced by the motion of basal/pyramidal interfaces – i.e. that the material deforms by the motion of these rather unusual interfaces which corresponds to a sort of "mechanical twinning without twinning orientation relationship and twinning plane". The reported phenomenon is interesting, the manuscript is very well written and would be of interest to the wide research community.

Comments

1. Although in-situ nanopillar, it is still single crystal deformation experiment, and as such, it shall contain analysis of resolved shear stresses into possible slip and twinning systems possibly activated in compression og Mg single crystal. This is even more urgent since the manuscript is intended for large audience beyond the magnesium community.

2. Many experiments in the literature seem to suggest that deformation twinning is suppressed in nanopillar experiments but other reports show deformation twinning in micropillar experiments at 100 MPa level. I understand that it depends on the orientation and sense of loading but still, can you comment how does it apply to Mg compressed along the c-axis?

3. Although it seems from the video that plastic deformation was mainly mediated by dislocation slip, as the authors claim, this referee is curious whether deformation twinning was activated in the experiment? Are the load drops observed on all stress-strain curves due to deformation twinning?

4. Looking at the video and figure 2, I am a bit confused. According to the experimental points in figures, the sudden deformation (called abrupt pancaking by authors) occurs extremely fast (in final two data points only) and it is accompanied by the load drop. This is the case, in spite of the fact that the pillar cross section area sharply increases (stress drop is even sharper). The deformation is stopped then and the pillar is unloaded. Why? The title of the work is "Rejuvenation of plasticity by deformation graining in magnesium". Since the renewal of dislocation plasticity after deformation graining is claimed to be the main result, the reader would like to have information about the rejuvenated dislocation slip after the event. Although the authors claim on the page 2 "After this dramatic strain burst, further height reduction would be possible, if the compressing was continued", they did not continue the loading, why?

5. Based on the results presented in figures 3 and 4, the reader is told that the deformation graining process (reversible on loading/unloading) takes place via reversible motion of pyramidal/ basal plane interface that changes the orientation of the crystal lattice reversibly. On the other hand, the sudden event changes and refines the crystal lattice orientations extremely fast (probably because dislocation slip is heavily involved in this sudden deformation) and irreversibly. A question what the authors call deformation graining? Is it the former reversible process or the latter irreversible process?

6. If a single crystal deforms by deformation twinning or martensitic transformation (i.e. deformation is localized into a mobile interface), strain compatibility at the mobile interface must be fulfilled. This has to be the case also in this experiment (here I assume that the deformation graining is the former reversible process). To assure the strain compatibility at the mobile interface, it must be inclined 50-60° to the compression axis (in fact figure 3 seems to suggest that). But the pillar is 3D single crystal object, the interface must be inclined while viewed from the side as well. How is this fulfilled in 3D pillar from the point of view of single crystal deformation crystallography?

7. Considering the “deformation graining” to be the former reversible process, resulting in rotation of the crystal lattice, the compressive strain can be calculated from lattice parameters (Fig. 7c), as the authors did. But the calculated value 5.85% is very low compared to the experimentally measured values of ~20% strain

8. On the other hand, considering the “deformation graining” to be the latter irreversible process resulting in the refinement of the single crystal via formation of various new interfaces and multiple lattice rotations, this deformation can be possibly larger. Can you calculate (estimate) this strain theoretically (based on the TEM results) and compare the obtained value with the experimental measured strain during the sudden event.

9. Is the deformation of the pillar transverse isotropic? Can we see the top view of the deformed single crystal? All grains are reported to be created by rotation within $\langle 2-1-10 \rangle$ single crystal zone, which implicitly means that the strain in this direction is zero. Is that true?

10. The authors carefully analyzed the grain orientations and interfaces, but they did not analyze the observed slip dislocations and possibly deformation twins formed before the sudden event. Why? They are probably very important.

11. Figure 5 (and S2-S5) suggest that the sudden event covers multiple deformation mechanisms giving rise to multiple new interfaces. Is it possible that the deformation graining (introduced as a new deformation mechanism proceeding via moving Py/B interface) has to be in fact coordinated with homogeneous shear of the lattice via dislocation slip and/or deformation twinning processes?

12. The MD simulation is also a bit unclear to me. Was the dislocation slip prohibited in the simulation? How comes that the deformation graining via the Py/B interfaces is preferred over deformation twinning in the simulation?

13. The deformation mechanisms in Mg and Mg alloys have been investigated in the literature for several decades. How comes that, except of the reference [22] which reports on a similar phenomenon in nanopillar experiment in compression on Mg single crystal along different axis, no one has ever seen such deformation mechanism to be activated in Mg single crystal experiment before? Is it an artifact of pillar experimentation?

14. Page 6 “the basal/prismatic interface migration (previously reported for Mg under c-axis extension) 22,23” the work is in many respects similar to the work reported in [22]. The difference is only in the compression loading axis orientation. Would it be possible to discuss a bit more the effect of load axis orientation and sense of load dependence of the new mechanism.

15. Page 6 “We expect the actual grain nucleation and growth in many situations to be a mixture of the two, i.e. hybrid diffusive-displacive.” ... this sentence is unclear to me

16. Page 7 “Once new grains form, the nearly exhausted dislocation plasticity of the heavily deformed single crystal is rejuvenated (as suggested in Figure 1d).” no, fig. 1d is still taken before the sudden event (abrupt pancaking)

17. Page 7 „This happens because the new grains have a low dislocation density. On the other hand, the moving boundaries may significantly reduce the dislocation density in the heavily deformed matrix (as suggested in Figure 1c), possibly by absorbing dislocations ahead of them.”No, processes like deformation twinning or martensitic transformation generally retain dislocation defects in the product lattice. Reference [26] reports atomistic simulation only. This does not mean that I want to say that GBs cannot act as dislocation sinks.

18. Concerning the mechanism of deformation graining, the explanation on page 5 “The mobility of Py/B interfaces, on the other hand, explains well the observed back and forth migration of the boundary during loading and unloading (Figure 3). Such stress-driven motion belongs to the “military transformation” category, the same as martensitic transformations, deformation twinning, etc., which may involve both affine shear and non-affine atomic shuffling components¹⁶.”is by far not sufficient to understand the new deformation process (moreover, here the authors again claim the former reversible process to be the deformation graining ?)

19. The idea that the lattice is reoriented by deformation twinning and refined by additional plasticity with the twin band is not new. Several articles (e.g. ref. [28]) reported this before. This shall be clearly referenced

20. Page 7-8 Nevertheless, the suggestion that the deformation graining mechanism possibly takes place during cold working of magnesium and leads to microstructure refinement and superformability is very attractive indeed.

21. Page 8 „In summary, the present work reveals a new low-temperature plasticity mode – “deformation graining”, which differs from deformation twinning and dislocation slip.“ since the mechanism of the deformation graining was not satisfactorily explained, this referee suggests that this sentence shall be avoided .

22. Page 8 „extent the DG mechanism can be active in bulk materials“ can be activated

23. The final sentence “In general, while it remains to be seen to what extent the DG mechanism can be active in bulk materials processing, our results have a major implication: if Mg (and possibly other HCP metals) is processed under high stress conditions (such as in samples of small dimensions or made of nanocrystalline grains, or under high strain rates like fast rolling or impact conditions), the normally poor c axis plasticity could be overturned through the activation of additional plastic deformation modes, which revive dislocation activities that are otherwise exhausted, even when the imposed loading starts out in the hardest orientation..“ is a very nice conclusion, although serious doubts remain....

24. The really important figures appear in the supplement, that is ridiculous.

25. This referee thinks that the deformation graining (abrupt pancaking in author’s terms), if it is real, cannot take place without previous and simultaneous dislocation slip and deformation twinning and hence it is irreversible deformation process in nature.

Responses to comments of reviewers

Reviewer #1

In this work, the authors investigated the deformation behavior of pure Mg micropillars by in-situ TEM analysis and MD simulation. The authors found a huge burst in on the stress-strain curves with an interesting pancaked deformation on the specimen geometry. Through the in-situ TEM observation and MD simulation, deformation graining by rearranging of atoms on the pyramidal plane to the basal plane, which contributes to generating new grains, is proposed. This mechanism contributes be to increasing the ductility of the material by decreasing the previous dislocations and activation of new slips in newly generated grains. It is a novel and interesting topic.

After reading the manuscript carefully, in the reviewer's opinion, the outcomes in the current work show sufficient novelty and the scientific clarity to merit publication in Nature communication after a revision. Given below are concerns and questions raised with respect to the key findings in the work.

Response: We appreciate the encouraging comments.

1) As the authors mentioned it is not possible to catch the deformation in stage II due to the time resolution limits in this stage. The reviewer suggests showing the orientation of the 'matrix' at the end of the homogeneous deformation (e.g. at point d in Figure 1 on the strain-stress curve) in order to exclude the possibility of double twinning and other possible twinning mechanisms.

Response: Thanks for the reviewer's suggestion. Figure 4 provides the orientation relationship. In the compression test of this pillar, we unloaded once the new grain emerged. The strain achieved before unloading is ~25.6%. Comparing to the strain achieved before burst of other tests (22%~39%, Table S2), we can infer that this pillar was near the end of the homogeneous deformation (note that it is very difficult to stop exactly at the point before burst because the burst happens very suddenly). The orientation relationship between the matrix and new grain of this pillar is $\sim 62^\circ \langle 2\bar{1}\bar{1}0 \rangle$ (Figure 4a). It is neither twinning nor double twinning relationship. We added the mechanical data of this pillar as P10 to the revised Figure S1 and Table S2. The main text is also revised accordingly (line 93).

Figure R1. The stress-strain curve of the pillar used in Figure 4. Insets, TEM bright field images before and after test. This figure was added into the revised Figure S1.

2) In the reviewer's opinion, the nucleation mechanism of the Pyr/B transformation or the new grains remains unclear. According to the supplementary video, between b and c (Figure 1), the total dislocation density decreased sharply based on the contrast. Either the dislocations escape from the surface or dynamic recovery occurs simultaneously. Therefore, again the reviewer recommends the authors investigate the sample before pancaking in detail, including the dislocation density distributions, dislocation type and the orientation of the material. In this way, we could have a clear understanding of the generation mechanism of those new grains better.

Response: Thanks for the reviewer's suggestion. The dislocation structures before pancaking can be referred to the work of *Ref. 11 (Science 365, 73-75, 2019)*, as emphasized in the revised manuscript (lines 33-36). In that work, *c*-axis compression was performed on Mg single crystal pillars (the same loading condition to the present work). Many pillars were compressed to a strain of 20%~30%, which is near the end of stage-1 deformation. High density $\langle c+a \rangle$ dislocations are generated in the entire pillar (Figure R2). These $\langle c+a \rangle$ dislocations are of edge, screw or mixed types. Their slip planes are $\{10\bar{1}1\}$ or $\{11\bar{2}2\}$, and they can cross-slip. Since the proposed pyramidal-basal transformation rearranges atoms from pyramidal to basal planes, the high density of dislocations on pyramidal planes may facilitate such transformation, probably by displacing the atomic positions on $\{10\text{-}11\}$ planes or by providing nucleation sites. We added the above discussion in the revised manuscript (line 120)

Figure R2. In situ TEM compression test showing that $\langle c+a \rangle$ dislocation slip mediates the plastic deformation of Mg pillar under c-axis compression (Fig. 1 in Ref. 11)

Regarding the likely decrease of dislocation density between b and c (Figure 1), it could be due to the slight tilting of the pillar during compression. Since the punch surface and the pillar top surface are generally not perfectly parallel to each other, the pillar could possibly be tilted slightly during loading. The dislocation contrast is very sensitive to the angle between electron beam and the sample. We did an experiment to show such effect (Figure R3). It is an as-deformed Mg sample with high density of dislocations inside. When we tilt the TEM holder by merely 0.5° (from -6.5° to -7.0°), the contrast of dislocations significantly changes. Actually, the dislocation density in Stage-1 increases continuously, as can be illustrated by Figure R2 and other results in Ref. 10.

Figure R3. An as-deformed Mg sample showing very different contrast when tilting merely 0.5° . (a) Tilt angle is -6.5° . The contrast of dislocations in areas 1 and 4 are obvious, but in areas 2 and 3 are very weak. (b) Tilt angle is -7.0° . The contrast of dislocations in areas 1 and 4 become weak, but in areas 2 and 3 are visible.

3) In Figure 3, to exclude the possibility of twinning and detwinning, the orientation of the new grains should be also given.

Response: The new grain in Figure 3 formed during compression. But we could not record the in-situ movie and diffraction patterns simultaneously, so we didn't characterize the orientation of this grain. The pillar then pancaked and became polycrystalline, so we could not identify where this grain comes from. That's why we carried out the interrupted experiment: we terminate compression and retracted the punch when a feature of new grain emerged in the view. The orientation is shown in Figure 4, which can exclude the possibility of any twinning system.

4) In Figures 2 and 5, according to the authors' scenario (Page 5 Line 4-18), the matrix will firstly transform to an orientation of grain 3 by Pyr/B and then twinned to grain 10. This indicates that the matrix should firstly be transformed to grain 3 nearly through the whole specimen dimension and then more than 80% of it twinned to grain 10. Is the Pyr/B transformation possible to happen all over the specimen? In addition, since the CRSS of tension twin is so low in Mg, under so high stress why grain 3 is not totally twinned? The reviewer is not convinced. Further explanations are necessary.

Response: We appreciate this comment. The possibility that the whole specimen firstly transformed to grain 3 which then twinned to grain 10 is low. The hardened specimen already has a very high dislocation density which would make Py-B transformation all over the specimen difficult due to possible interaction with the existing dislocations. Most likely, the Py-B transformation occurred in multiple locations in an explosive manner. If the specimen were continuously compressed after pancaking, it could be anticipated that grain 3 would totally twinned to grain 10. But this twinning process could be disrupted due to the abrupt reduction in the height of the specimen, leaving a residual portion of grain 3 in the specimen.

Figure R4. Two low angle grain boundaries within grain 10. (Figure S3b in the revise manuscript)

Also, by carefully examining the high-resolution TEM images acquired near grain 10, we found two low angle grain boundaries within grain 10. This suggests that the large grain 10 may be composed of three small grains that underwent same transformation route and merged together. The three possible sub-grains are marked by 10-a, 10-b and 10-c, as shown in Figure R4. The main text is also revised accordingly (line 65-67).

Figure R5. An example that a twin spreads all over the pillar within a single strain burst. The burst occurs within 0.1s (the recording rate is 10 frames/s).

Regarding the question whether Py/B transformation can happen all over the sample, we can have some clues from tension twin, which can be a special case of deformation induced “grain” formation. We have observed many times that tension twin can spread all over the pillar very fast (Figure R5) when the punch surface and pillar top surface is aligned well, which makes the stress concentration at the contact interface less significant (twin usually nucleates at such place with local high stress). As a result, twin nucleation will require a high applied load, and the stored elastic energy in the pillar is very high. Once a twin forms, it can grow very fast and spread all over the pillar. Presumably, Py/B transformation can also happen all over the specimen when the applied load is high enough. Regarding the question why grain 3 is not totally twinned, this could be due to the local stress distribution. As shown in Figure R5, although the applied load is as high as 0.65 GPa, there is still some untwinned places (red arrow).

5) Page 8 Line 4, the author cited as an example of high ductility in Mg single crystal at room temperature (Ref. 28). By carefully read this paper, from the reviewer’s understanding, the author compressed the crystal along $\langle 11\bar{2}0 \rangle$ direction, where tension twin is preferential and the new grains come out mostly by dynamic recrystallization at room temperature. This example here could probably mislead the readers about the difference between DG, twinning and DRX.

Response: Thank for this comment. The reason we cited this reference is that it shows formation of new grains at room temperature, while DRX typically occurs at elevated temperature. In order to avoid any possible misleading, we removed this sentence in the revised manuscript.

6) It would be very interesting if we could find this new mechanism in the bulk material since the size effect plays a crucial role in the mechanical response in such a small scale.

Response: We agree with the reviewer. It is our hope that this paper will stimulate new interest to look into grain boundary activities in Mg. In general, dislocation slip and deformation twinning are the main plastic deformation modes in bulk materials (coarse-grained) at room temperature. However, as the sample/grain size reduces, the critical stresses required for dislocation slip and twinning significantly increases. As a result, the grain boundary migration mediated plasticity, such as the Py/B transformation (and other potential modes that have yet been explored), may become significant enough and could play a crucial role in the mechanical response.

There are some hints in literatures that DG may play a role in microstructure evolution. P. Cizek and M. R. Barnett found $44^\circ\langle 1\bar{2}10 \rangle$ and $49^\circ\langle 1\bar{2}10 \rangle$ misorientations in deformed Mg alloy. The authors point out that these misorientations “cannot be unambiguously classified in terms of known twinning misorientations”^[1]. These two misorientations can neither be simply attributed to Py/B transformation. So there may be some other DG mode or complex multiple DG-twinning transformation. Further studies are needed to detect DG in bulk materials, as well as potential new DG mode. We added the above discussion in the revised manuscript (line 247).

[1] P. Cizek and M. R. Barnett, Scripta Materialia, 59, 959-962, 2008

Reviewer #2

The manuscript present results of nanopillar experiments on Mg single crystal deformed along the c-axis in compression at room temperature. As the pillar is compressed, plastic flow due to non basal dislocation slip (clearly visible on the video) gradually strain hardens the single crystal up to gigapascal level, at which new grain is formed and then the crystal suddenly deforms plastically ~20%, while the applied load decreases without fracture. The accompanying video very nicely documents this unusual deformation process. After this large compressive deformation and unloading, the single crystal becomes decomposed into multiple small grains that share a common rotation a-axis of the HCP lattice. The new microstructure consisting of these multiple small grains within the deformed single crystal pillar is carefully analyzed by TEM a HRTEM as concerns the lattice rotation and type of new interfaces. The deformation process is claimed to represent new deformation mechanism of magnesium called “deformation graining” activated at low temperatures and high stresses. It is suggested that this new deformation mechanism can be possibly activated in cold working of magnesium.

The authors claim that the observed lattice rotations are produced by the motion of basal/pyramidal interfaces – i.e. that the material deforms by the motion of these rather unusual interfaces which corresponds to a sort of “mechanical twinning without twinning orientation relationship and twinning plane”. The reported phenomenon is interesting, the manuscript is very well written and would be of interest to the wide research community.

Response: We appreciate the encouraging comments.

1. Although in-situ nanopillar, it is still single crystal deformation experiment, and as such, it shall contain analysis of resolved shear stresses into possible slip and twinning systems possibly activated in compression on Mg single crystal. This is even more urgent since the manuscript is intended for large audience beyond the magnesium community.

Response: Thanks for the reviewer’s suggestion. The Schmid factors of all slip and twinning systems under c-axis compression are listed in the table below. The possibly activated deformation modes are <c+a> slip and contraction twinning. The table is added in the revised manuscript (Table S1 and line 30).

Slip and twinning systems	Schmid factors	CRSS in the yield point in Figure 1g ($\sigma_y=680 \text{ MPa}$)
$\{0002\}\langle 2\bar{1}\bar{1}0\rangle$ (basal $\langle a \rangle$ slip)	0	-
$\{10\bar{1}0\}\langle 2\bar{1}\bar{1}0\rangle$ (prismatic $\langle a \rangle$ slip)	0	-
$\{10\bar{1}0\}\langle 0001\rangle$ (prismatic $\langle c \rangle$ slip)	0	-
$\{10\bar{1}1\}\langle 2\bar{1}\bar{1}0\rangle$ (pyramidal I $\langle a \rangle$ slip)	0	-
$\{10\bar{1}1\}\langle 2\bar{1}\bar{1}3\rangle$ (pyramidal I $\langle c+a \rangle$ slip)	0.4	272
$\{11\bar{2}2\}\langle 2\bar{1}\bar{1}3\rangle$ (pyramidal II $\langle c+a \rangle$ slip)	0.45	306
$\{10\bar{1}1\}\langle 10\bar{1}\bar{2}\rangle$ (contraction twinning)	0.41	-
$\{10\bar{1}2\}\langle 10\bar{1}\bar{1}\rangle$ (extension twinning)	-0.5	-

2. Many experiments in the literature seem to suggest that deformation twinning is suppressed in nanopillar experiments but other reports show deformation twinning in micropillar experiments at 100 MPa level. I understand that it depends on the orientation and sense of loading but still, can you comment how does it apply to Mg compressed along the c -axis?

Response: Thanks for the reviewer's suggestion. Deformation twinning is very sensitive to grain size ^[1]. When grain size or sample size decrease to micro or nano scale, the nucleation stress for deformation twinning will dramatically increase. For $\{10\bar{1}2\}$ extension twin, which is activated under c -axis tension, the CRSS in bulk scale Mg single crystal is of the order of 1.0 MPa ^[2], while it increases to the order of 10 MPa in microscale single crystal ^[3] and to the order of 100 MPa in submicron scale ^[4]. For $\{10\bar{1}1\}$ contraction twin, it is activated under c -axis compression. It was found to be suppressed in micro scale single crystal ^[5,6] and submicron scale single crystal ^[7], while pyramidal slip dominates the plastic deformation ^[7], indicating that contraction twinning is more sensitive to sample size than dislocation slip in HCP Mg, which is similar to the case in HCP Ti alloy under c -axis compression ^[1]. We pointed out such size effect (line 233) and orientation effect (line 211) on twinning in the revised manuscript.

[1] Yu et al., Nature, 463, 7279, 335-338 (2010)

[2] Yu et al., Philosophical Magazine Letters, 91, 12, 757-756 (2011)

[3] Liu et al., Acta Materialia, 135, 411-421, (2017)

[4] Liu et al., Nature Communications, 5, 3297, (2014)

[5] Lilleodden, Scripta Materialia, 62, 8, 532-535 (2010)

[6] Byer et al., Scripta Materialia, 62, 8, 536-539 (2010)

[7] Liu et al., Science 365, 73-75, (2019)

3. Although it seems from the video that plastic deformation was mainly mediated by dislocation slip, as the authors claim, this referee is curious whether deformation twinning was activated in the in the experiment? Are the load drops observed on all stress-strain curves due to deformation twinning?

Response: By carefully examining the in-situ video and post mortem images, it can be confirmed that no deformation twinning occurs in stage-1 deformation in the present work. The load drops correspond to the dislocation activities, e.g. fast formation and propagation. We added such description in the revised manuscript (line 43).

4. Looking at the video and figure 2, I am a bit confused. According to the experimental points in figures, the sudden deformation (called abrupt pancaking by authors) occurs extremely fast (in final two data points only) and it is accompanied by the load drop. This is the case, in spite of the fact that the pillar cross section area sharply increases (stress drop is even sharper). The deformation is stopped then and the pillar is unloaded. Why? The title of the work is “Rejuvenation of plasticity by deformation graining in magnesium”. Since the renewal of dislocation plasticity after deformation graining is claimed to be the main result, the reader would like to have information about the rejuvenated dislocation slip after the event. Although the authors claim on the page 2 “After this dramatic strain burst, further height reduction would be possible, if the compressing was continued”, they did not continue the loading, why?

Response: There are two reasons we didn't perform further compression. (1) We took the pancaked samples off from the loading device for TEM characterization. Some of them were thinned down to 100~200 nm for HRTEM, such thin foil is no longer suitable for mechanical loading. (2) A pancaked sample has a larger top surface thus a larger adhesive force with the punch. So, the sample will be pulled apart and sticks on the punch during unloading. The contaminated punch can no longer be used, unless a new one is purchased (~\$2000) or cleaned by FIB (time consuming and costly). This problem will become worse if we continuously compress the pancaked sample.

In the first version of the manuscript, although no further compression was performed, we are able to deduce that the plasticity is rejuvenated. First, the dislocation plasticity was nearly exhausted at the end of stage-1 deformation; second, in the new grains, high density basal dislocations are found, as well as extension twins, which should be the

evidence for continuous plasticity, because before new grain formation, the compression axis is c -axis, basal slip and $\{10\bar{1}2\}$ twinning should not occur.

In order to verify the compressive deformability after pancaking, we conducted three additional experiments. One typical example is shown in Figure R6. The other two exhibited similar phenomena (not shown here). During the first compression, the pillar exhibited similar two stages deformation like other pillars. During further compression, the pancaked sample continuously flattened. When the flattened sample reaches a size comparable with that of the substrate, the stress-strain curve comprises the deformation of the substrate. When we retracted the punch, the sample was firmly adhered to the punch. At last, part of the substrate was pulled apart and stick on the punch. We added this result to the revised manuscript (line 52 and P11 in Figure S1 and table S2).

Figure R6. An example of a pancaked sample that underwent continuous compression. (a) Corresponding engineering stress-strain curve. (b) The flattened sample and part of the substrate are pulled apart and stick on the diamond punch.

5. Based on the results presented in figures 3 and 4, the reader is told that the deformation graining process (reversible on loading/unloading) takes place via reversible motion of pyramidal/ basal plane interface that changes the orientation of the crystal lattice reversibly. On the other hand, the sudden event changes and refines the crystal lattice orientations extremely fast (probably because dislocation slip is heavily involved in this sudden deformation) and irreversibly. A question what the authors call deformation graining? Is it the former reversible process or the latter irreversible process?

Response: Deformation graining is a process that new grains form and grow from the matrix by mechanical loading. The Py-B transformation is reversible, which is a

mechanism that controls the grain nucleation and boundary motion. But the total shape-change and microstructure evolution caused by DG and subsequent dislocation and twinning activities are irreversible. This can be an analogy with deformation twinning: twin boundary migration is reversible, but the resultant microstructure by twinning, slip and multiple twinning cannot be recovered. We clarified this in the revised manuscript (line 209).

6. If a single crystal deforms by deformation twinning or martensitic transformation (i.e. deformation is localized into a mobile interface), strain compatibility at the mobile interface must be fulfilled. This has to be the case also in this experiment (here I assume that the deformation graining is the former reversible process). To assure the strain compatibility at the mobile interface, it must be inclined 50-60° to the compression axis (in fact figure 3 seems to suggest that). But the pillar is 3D single crystal object, the interface must be inclined while viewed from the side as well. How is this fulfilled in 3D pillar from the point of view of single crystal deformation crystallography?

Response: This is a great question. For crystals with cubic structures, a mobile interface should be inclined 50~60° to the compression axis, such that the displacement field generated by the shear deformation at the interface is able to fulfill the strain compatibility. In HCP single crystals, the scenario is more complex. For example, for {10-12} twin boundary, it has high mobility even when it is perpendicular or parallel to the tensile axis along the c-axis because in this orientation, the lattice transformations, i.e. basal becomes prismatic and prismatic becomes basal, can be achieved readily. The so-called basal/prismatic (BP) interfaces has been observed in extensive experiments and simulations. For a pure BP interface, it can migrate easily even when it is perpendicular to the tension or compression axis (inclined 90°). The strain compatibility is achieved by the misfit strains produced by the lattice reorientation (by ~90° around $\langle 2\bar{1}\bar{1}0 \rangle$). Along the zone axis of the twins, the strain is zero.

In the case of Py-B transformation, the interplanar spacing decreases, so Py-B transformation is favored in the c-axis compression. Similarly, along the zone axis of new grain produced by Py-B transformation, i.e. $\langle 2\bar{1}\bar{1}0 \rangle$, the strain is zero (i.e. when viewed from the side), as shown in the response of question #9. In general, the local strain produced by lattice reorientation plays an important role in strain accommodation in HCP crystals.

7. Considering the “deformation graining” to be the former reversible process, resulting in rotation of the crystal lattice, the compressive strain can be calculated from lattice parameters (Fig. 7c), as the authors did. But the calculated value 5.85% is very low compared to the experimentally measured values of ~20% strain.

Response: Thanks for pointing this out. The total strain produced in Stage-2 comprises two contributions: the strain produced by Py-B transformation (~5.85%), and the subsequent dislocation slip and deformation twinning in the new grains. Considering that the strain produced by Py-B transformation is relatively small, the strain produced by dislocation and twinning in the new grains should be the main contributor to the stage-2 deformation. We analyzed it in the revised manuscript (line 224). The contribution of each plastic deformation modes is estimated in the revised manuscript (Table S4), as also can be seen in the answer to the next question.

8. On the other hand, considering the “deformation graining” to be the latter irreversible process resulting in the refinement of the single crystal via formation of various new interfaces and multiple lattice rotations, this deformation can be possibly larger. Can you calculate (estimate) this strain theoretically (based on the TEM results) and compare the obtained value with the experimental measured strain during the sudden event.

Response: We tried to estimate the strain produced by Py-B transformation, twinning and dislocation slip based on TEM image in the revised manuscript (Table S4).

The contribution of ε_{Py-B} , ε_{twin} and $\varepsilon_{dislocation}$ in the strain produced in Stage-2 ($\varepsilon_{Stage-2}$) can be estimated by: (1) $\varepsilon_{Py-B} = f_1 \times 5.85\%$, (2) $\varepsilon_{twin} = f_2 \times 6.28\%$, (3) $\varepsilon_{dislocation} = \varepsilon_{Stage-2} - \sum(\varepsilon_{Py-B} + \varepsilon_{twin})$, where f_1 is the volume fraction of the grains that experienced Py-B transformation, and f_2 the volume fraction of $\{10\bar{1}2\}$ twins, 5.85% and 6.28% are the plastic strains produced by Py-B transformation and $\{10\bar{1}2\}$ twinning, respectively. Here we chose the pillar shown in Figure 2 as an example. Using the above method, the total strain mediated by Py-B transformation, $\{10\bar{1}2\}$ twinning and dislocation slip is 3.85%, 3.85% and 12%, respectively.

9. Is the deformation of the pillar transverse isotropic? Can we see the top view of the deformed single crystal? All grains are reported to be created by rotation within $\langle 2-1-$

10> single crystal zone, which implicitly means that the strain in this direction is zero. Is that true?

Response: The reviewer is correct. When the grains “rotate” around $\langle 2\bar{1}\bar{1}0 \rangle$, the deformation along the thickness direction $\langle 2\bar{1}\bar{1}0 \rangle$ should be small, as shown in Figure R7. We added this into the revised Figure S1 (P8).

Figure R7. Top view images of the P8 pillar. (a) Before compression. (b) After compression.

10. The authors carefully analyzed the grain orientations and interfaces, but they did not analyze the observed slip dislocations and possibly deformation twins formed before the sudden event. Why? They are probably very important.

Response: The dislocation structures in stage-1 deformation have been systematically investigated by the work of *Ref. 11*. In that work, *c*-axis compression was performed on Mg single crystal pillars. Many pillars were compressed to a strain of 20%~30%, which is near the end of stage-1 deformation. High density $\langle c+a \rangle$ dislocations are generated in the entire pillar (Figure R8).

Figure R8. In situ TEM compression test showing that $\langle c+a \rangle$ dislocation slip mediates the plastic deformation of Mg pillar under *c*-axis compression (Fig. 1 in Ref. 11)

These $\langle c+a \rangle$ dislocations are of edge, screw or mixed types. Their slip planes are $\{10\bar{1}1\}$ or $\{11\bar{2}2\}$, and they can cross-slip. Since the proposed pyramidal-basal

transformation rearranges atoms from pyramidal to basal planes, the high density of dislocations on pyramidal planes may facilitate such transformation, probably by displacing the atomic positions on $\{10\bar{1}1\}$ planes or by providing nucleation sites. We added the above discussion in the revised manuscript (lines 33-36, 120).

11. Figure 5 (and S2-S5) suggest that the sudden event covers multiple deformation mechanisms giving rise to multiple new interfaces. Is it possible that the deformation graining (introduced as a new deformation mechanism proceeding via moving Py/B interface) has to be in fact coordinated with homogeneous shear of the lattice via dislocation slip and/or deformation twinning processes?

Response: At the early stage of DG when the new grains do not impinge, migration of Py/B interfaces may not need to be coordinated with dislocation slip or deformation twinning. For example, as shown in Figure 5, there is neither dislocation nor twin within the new grain. The observed dislocation slip and deformation twinning are subsequent processes after DG. When the grains impinge on each other, further boundary migration will need dislocation slip to satisfy strain compatibility at grain boundaries. It's also worth noting that DG cannot accommodate the strain along the "rotation" axis, just as twinning cannot accommodate strains along the zone axis, so dislocation slip is needed to make the deformation more homogeneous. We addressed this comment in the revised manuscript (line 229).

12. The MD simulation is also a bit unclear to me. Was the dislocation slip prohibited in the simulation? How comes that the deformation graining via the Py/B interfaces is preferred over deformation twinning in the simulation?

Response: We thank the reviewer for the good question. We did not prohibit dislocation slip in the simulation and only applied *c*-axis compression on the single crystal. We used the EAM potential developed by Sun et al. (Ref. 40). After we carefully checked the simulation results, we found that $\{10\bar{1}1\}$ contraction twinning actually occurred at the very beginning of plastic deformation. But the twin embryos only grew a few nanometers wide, then new grains with Py/B interfaces formed almost immediately near the twins. The new grains grew and quickly consumed the contraction twins. A possibility is that, growth of contraction twins requires zonal twinning dislocations that comprise two or four consecutive $\{10\bar{1}1\}$ planes. Nucleation and glide of zonal

twinning dislocations is not easy. Under high stresses in the submicron scale, forming the mobile Py/B interfaces is an alternative mode to accommodate the plastic strain.

13. The deformation mechanisms in Mg and Mg alloys have been investigated in the literature for several decades. How comes that, except of the reference [22] which reports on a similar phenomenon in nanopillar experiment in compression on Mg single crystal along different axis, no one has ever seen such deformation mechanism to be activated in Mg single crystal experiment before? Is it an artifact of pillar experimentation?

Response: There may be two reasons why this mechanism has not been reported before: (1) The DG observed in the present work occurred under *c*-axis compression at a high stress level of ~1.0 GPa. Such stress level is hard to be achieved in bulk single crystal Mg. (2) Although the deformation mechanisms in Mg have been investigated for decades, however, to the best of the authors' knowledge, only limited works have conducted *c*-axis compression on Mg micro/nanopillars [1-3]. These few studies focus on the dislocation behaviors during the early deformation, and the performed strain was at the level of about ~10%. In contrast, DG observed in the present work occurred at the strain of 20%~30%. Moreover, BP interfaces associated with the mechanism reported in Ref. 22 (in the revised manuscript the ref number is 23) have been widely observed in bulk scale Mg alloys, not only in nanopillars, so we expect that the new mechanism reported here may potentially have a more general effect in bulk scale.

[1] Lilleodden, Scripta Materialia, 62, 8, 532-535 (2010)

[2] Byer et al., Scripta Materialia, 62, 8, 536-539 (2010)

[3] Zhang et al., Journal of Materials Research, 34, 9, 1542-1554 (2019)

14. Page 6 “the basal/prismatic interface migration (previously reported for Mg under c-axis extension) 22,23“ the work is in many respects similar to the work reported in [22]. The difference is only in the compression loading axis orientation. Would it be possible to discuss a bit more the effect of load axis orientation and sense of load dependence of the new mechanism.

Response: Thanks for the reviewer's suggestion. The interplanar spacing of pyramidal $\{10\bar{1}1\}$ equals 0.245 nm, and the interplanar spacing of basal plane equals 0.261 nm.

Thus, the Py-B transformation reduces the interplanar spacing along the c-axis of the matrix crystal. This explains why the Py-B transformation is favored in c-axis compression. The interplanar spacing of prismatic $\{10\bar{1}0\}$ equals 0.278 nm, thus the basal/prismatic transformation increases the interplanar spacing in c-axis tension. We added this discussion in the revised manuscript (line 211)

15. Page 6 “We expect the actual grain nucleation and growth in many situations to be a mixture of the two, i.e. hybrid diffusive-displacive.” ... this sentence is unclear to me

Response: In the revised manuscript, we added one paragraph to broaden the discussion of deformation induced grain formation and growth. (lines 183-199).

“In the present work, the grain formation and boundary migration are accomplished through the mechanical-loading induced Py-B transformation (“military” displacement of atoms from one crystallographic plane to another). However, it is worth noting that, in a more general case, mechanical loading not only produces affine displacements of atoms, but also can produce non-affine displacements. These non-affine atomic displacements consist of domain-uniform “shuffling” that is of identical pattern from unit-cell to unit-cell (in the same transformation domain), and may plus individually randomized motion (analogy with “civilian” movement). The latter, which is called deformation-induced diffusion^[1], has been found to contribute to the crystallization of metallic glasses under cyclic loading well below the glass transition temperature^[2]. At this point, while we can rigorously measure the affine transformation strain aspect, we cannot ascertain the ratio of domain-uniform “shuffling” versus “civilian” motion, but we expect that the actual grain nucleation and growth in many situations probably be a mixture of the two^[3], i.e. hybrid diffusive-displacive mode, at room temperature. We note that at the atomic level, the essential difference between a “civilian” and a “military” process is that, while the atomic registries (i.e. nearest-neighbor relationships) can change in both types of processes, changes in the former occur in a highly individually randomized manner, while in the latter in a much more collective, domain-uniform manner as reflected quantitatively by the larger activation volume^[4].”

[1] Li, W., Rieser, et al. Deformation-driven diffusion and plastic flow in amorphous granular pillars. Phys Rev E, 91, 062212, (2015).

[2] Wang, C.-C. et al. Real-time, high-resolution study of nanocrystallization and fatigue cracking in a cyclically strained metallic glass. PNAS, 110, 19725-19730,

(2013).

[3] Han, J., Thomas, S. L. & Srolovitz, D. J. Grain-boundary kinetics: A unified approach. *Prog. Mater. Sci.* 98, 386-476, (2018)

[4] Li, J. The Mechanics and Physics of Defect Nucleation. *MRS Bull.* 32, 151-159, (2011).

16. Page 7 “Once new grains form, the nearly exhausted dislocation plasticity of the heavily deformed single crystal is rejuvenated (as suggested in Figure 1d).” no, fig. 1d is still taken before the sudden event (abrupt pancaking)

Response: Although Figure 1d is still taken before burst, it shows dark contrast emerged and spread over the new grain, suggesting dislocation or twinning activity inside. Considering that we did not prove that dislocations or twins actually exist in the region marked in Figure 1d, in the revised manuscript, we use Figure 2 (high density dislocations inside the new grains) and Figure S1-P11 (large compressive deformability after pancaking) as the evidence of rejuvenated plasticity (line 222).

17. Page 7 “This happens because the new grains have a low dislocation density. On the other hand, the moving boundaries may significantly reduce the dislocation density in the heavily deformed matrix (as suggested in Figure 1c), possibly by absorbing dislocations ahead of them.”No, processes like deformation twinning or martensitic transformation generally retain dislocation defects in the product lattice. Reference [26] reports atomistic simulation only. This does not mean that I want to say that GBs cannot act as dislocation sinks.

Response: Thanks for pointing this out. To the authors’ knowledge, so far there is no experiment evidence that moving basal/prismatic and Py/B interfaces as well as TBs can absorb dislocations like GBs that serve as dislocation sink, so we remove this statement in the revised manuscript.

18. Concerning the mechanism of deformation graining, the explanation on page 5 “The mobility of Py/B interfaces, on the other hand, explains well the observed back and forth migration of the boundary during loading and unloading (Figure 3). Such stress-driven motion belongs to the “military transformation” category, the same as

martensitic transformations, deformation twinning, etc., which may involve both affine shear and non-affine atomic shuffling components¹⁶.”is by far not sufficient to understand the new deformation process (moreover, here the authors again claim the former reversible process to be the deformation graining?)

Response: Thanks for this comment. Deformation graining is a process that new grains form and grow from the matrix by mechanical loading. The Py-B transformation is a mechanism that controls the grain nucleation and boundary motion. The reversible process is strictly referred to the motion of the Py/B interfaces. But the total shape-change and microstructure evolution, which involves subsequent dislocation and twinning activities, are irreversible. This can be an analogy with deformation twinning: twin boundary migration is reversible, but the resultant microstructure by twinning, slip and multiple twinning is not reversible.

19. The idea that the lattice is reoriented by deformation twinning and refined by additional plasticity with the twin band is not new. Several articles (e.g. ref. [28]) reported this before. This shall be clearly referenced.

Response: Thanks for this suggestion. We have stated and referenced this effect in the revised manuscript (line 241).

20. Page 7-8 Nevertheless, the suggestion that the deformation graining mechanism possibly takes place during cold working of magnesium and leads to microstructure refinement and superformability is very attractive indeed.

Response: Thanks for this encouraging comment.

21. Page 8 “In summary, the present work reveals a new low-temperature plasticity mode –“deformation graining”, which differs from deformation twinning and dislocation slip.”..... since the mechanism of the deformation graining was not satisfactorily explained, this referee suggests that this sentence shall be avoided.

Response: This sentence is revised as “In summary, the present work reveals a new phenomenon, “deformation graining”, which has the potential of enhancing the plasticity of Mg under proper activating conditions.” (line 244).

22. Page 8 “*extent the DG mechanism can be active in bulk materials*“ *can be activated.*

Response: Thanks for the careful examination. We corrected this typo.

23. *The final sentence “In general, while it remains to be seen to what extent the DG mechanism can be active in bulk materials processing, our results have a major implication: if Mg (and possibly other HCP metals) is processed under high stress conditions (such as in samples of small dimensions or made of nanocrystalline grains, or under high strain rates like fast rolling or impact conditions), the normally poor c-axis plasticity could be overturned through the activation of additional plastic deformation modes, which revive dislocation activities that are otherwise exhausted, even when the imposed loading starts out in the hardest orientation..“ is a very nice conclusion, although serious doubts remain....*

Response: Thanks for this comment. The point we are trying to make is that, when the common deformation modes are nearly exhausted or difficult to be activated, DG may serve as an additional deformation mode to enable further plasticity under high stresses. We revised the sentences (line 252) to emphasize that such implication definitely requires further studies.

“Our results may also have another implication: if Mg (and possibly other HCP metals) is processed under high-stress conditions (such as in samples of small dimensions or made of nanocrystalline grains, or under high strain rates like fast rolling or impact conditions), the normally poor c-axis plasticity may be overturned through the activation of additional plastic deformation modes, which revive dislocation activities that are otherwise exhausted, even when the imposed loading starts out in the hardest orientation. In general, to what extent the DG mechanism can be activated in bulk materials, as well as its possible effects on mechanical properties, microstructure evolution and processability require further exploration.”

24. *The really important figures appear in the supplement, that is ridiculous.*

Response: We thought there was a limit on the number figures in the main text. In the revised version, we have moved the Figures S5 and S7 to the main text as Figures 5 and

6. In the new Figure 5, we added atomic scale images to show the fine structures of Py/B and $\{10\bar{1}0\}/\{10\bar{1}3\}$ interfaces of a nano-sized new grain formed during c-axis compression.

25. This referee thinks that the deformation graining (abrupt pancaking in author's terms), if it is real, cannot take place without previous and simultaneous dislocation slip and deformation twinning and hence it is irreversible deformation process in nature.

Response: Thanks for this comment. Formation of new grains and the subsequent pancaking in a heavily deformed single crystal Mg specimen as observed in this work is indeed not reversible, from the standpoint of thermodynamics. The migration of Py/B interfaces in the process of forming the new grains, as shown in our loading-unloading experiment, is reversible (line 209 in the revised manuscript). As discussed in the previous answers, this can be an analogy with deformation twinning: twin boundary migration is reversible, but the resultant microstructure by twinning, slip and multiple twinning is not reversible.

The high density of pyramidal dislocations may facilitate the new grain formation via Py-B transformation, probably by displacing the atoms on pyramidal planes or by providing nucleation sites. But whether the Py-B transformation should be preceded by dislocation slip, i.e. the atomistic mechanism that controls the grain nucleation (atom displacement pathway, shuffling, energy barrier ...), requires further studies.

REVIEWERS' COMMENTS

Reviewer #1 (Remarks to the Author):

The authors have responded admirably to the concerns of the reviewer. I have no further reservations.

Reviewer #2 (Remarks to the Author):

The authors responded to all referee comments and the referee is satisfied with the answers. The authors also included many amendments into revised version including new figures. In the opinion of the referee, these changes improved the manuscript and made it more easily understandable to wider community. The video S1 very nicely documents the claimed rejuvenation of dislocation plasticity. Although the referee still has some reservations about the provided explanation of the mechanism of the DG, main strength of the manuscript consists in that it reports on a novel deformation mechanism and provides motivation for future research.

Regarding the comment 9. The authors included the Top view of the sample into Fig. S1 (P8) but did not mention in the caption as well as in the text that the pancaking is highly transverse anisotropic. The referee considers this to be very important.

Responses to comments of reviewers

Reviewer #1

The authors have responded admirably to the concerns of the reviewer. I have no further reservations.

Response: Thanks for the reviewer.

Reviewer #2

The authors responded to all referee comments and the referee is satisfied with the answers. The authors also included many amendments into revised version including new figures. In the opinion of the referee, these changes improved the manuscript and made it more easily understandable to wider community. The video S1 very nicely documents the claimed rejuvenation of dislocation plasticity. Although the referee still has some reservations about the provided explanation of the mechanism of the DG, main strength of the manuscript consists in that it reports on a novel deformation mechanism and provides motivation for future research.

Response: Thanks for the reviewer.

Regarding the comment 9. The authors included the Top view of the sample into Fig. S1 (P8) but did not mention in the caption as well as in the text that the pancaking is highly transverse anisotropic. The referee considers this to be very important.

Response: Thanks for this suggestion. We mentioned this point in the revised main text and figure caption.

Page 9, line 252 in main text: “Moreover, in our experiments, formation of new grains by Py-B transformation “rotates” the crystal around $\langle 2\bar{1}\bar{1}0 \rangle$ but cannot provide strain along this “rotation” axis, that is, the deformation is anisotropic, as shown by the top view of the flattened sample P8 in Supplementary Figure 1. As a result, dislocation slip is needed to make the deformation more homogeneous.”

In the caption of Supplementary Figure 1: “The top view of the flattened sample P8 is provided, indicating that the pancaking is transverse anisotropic.”